# Neutralizing mAbs against SFTS Virus Gn Protein Show Strong Therapeutic Effects in an SFTS Animal Model

**DOI:** 10.3390/v14081665

**Published:** 2022-07-28

**Authors:** Masayuki Shimojima, Satoko Sugimoto, Kunihiko Umekita, Taishi Onodera, Kaori Sano, Hideki Tani, Yuki Takamatsu, Tomoki Yoshikawa, Takeshi Kurosu, Tadaki Suzuki, Yoshimasa Takahashi, Hideki Ebihara, Masayuki Saijo

**Affiliations:** 1Department of Virology I, National Institute of Infectious Diseases, Tokyo 208-0011, Japan; ssugimo@niid.go.jp (S.S.); yukiti@nagasaki-u.ac.jp (Y.T.); ytomoki@niid.go.jp (T.Y.); kurosu@niid.go.jp (T.K.); hebihara@niid.go.jp (H.E.); 2Department of Respirology, Rheumatology, Infectious Diseases and Neurology, Internal Medicine, Faculty of Medicine, University of Miyazaki, Miyazaki 889-1692, Japan; kunihiko_umekita@med.miyazaki-u.ac.jp; 3Research Center for Drug and Vaccine Development, National Institute of Infectious Diseases, Tokyo 162-8640, Japan; taishi@niid.go.jp (T.O.); ytakahas@niid.go.jp (Y.T.); 4Department of Pathology, National Institute of Infectious Diseases, Tokyo 162-8640, Japan; ka--sano@niid.go.jp (K.S.); tksuzuki@niid.go.jp (T.S.); 5Department of Virology, Toyama Institute of Health, Toyama 939-0363, Japan; toyamaeiken3@juno.ocn.ne.jp; 6Department of Virology, Institute of Tropical Medicine, Nagasaki University, Nagasaki 852-8523, Japan; 7Medical Affairs Department, Health and Welfare Bureau, Sapporo 060-0042, Japan

**Keywords:** severe fever with thrombocytopenia syndrome (SFTS), SFTS virus, therapeutic monoclonal antibody, neutralization, animal model

## Abstract

Severe fever with thrombocytopenia syndrome (SFTS) is an infectious disease with a high case fatality rate caused by the SFTS virus, and currently there are no approved specific treatments. Neutralizing monoclonal antibodies (mAbs) against the virus could be a therapeutic agent in SFTS treatment, but their development has not sufficiently been carried out. In the present study, mouse and human mAbs exposed to the viral envelope proteins Gn and Gc (16 clones each) were characterized in vitro and in vivo by using recombinant proteins, cell culture with viruses, and an SFTS animal model with IFNAR^-/-^ mice. Neutralization activities against the recombinant vesicular stomatitis virus bearing SFTS virus Gn/Gc as envelope proteins were observed with three anti-Gn and six anti-Gc mAbs. Therapeutic activities were observed among anti-Gn, but not anti-Gc mAbs with neutralizing activities. These results propose an effective strategy to obtain promising therapeutic mAb candidates for SFTS treatment, and a necessity to reveal precise roles of the SFTS virus Gn/Gc proteins.

## 1. Introduction

Severe fever with thrombocytopenia syndrome (SFTS) is an acute, infectious disease caused by the SFTS virus which was first reported by Chinese research groups in 2011 [1,2] and is currently endemic in China, Korea, and Japan. Case fatality rates (CFRs) of the disease reported are as high as 18% in China [3] and 27% in Japan [4]. Although there are currently no approved treatments specific to the disease, clinical studies with Avigan^®^ (FUJIFILM Toyama Chemical Co., Ltd., Tokyo, Japan), a nucleoside analogue anti-influenza drug in Japan, performed in China and Japan independently showed improvement of CFRs of SFTS patients [3,5,6]. Screening of an approved drug library found calcium channel blockers having inhibitory effects on SFTS virus growth and retrospective studies revealed that CFR within patients being treated with a calcium channel blocker nifedipine, one of the most widely used drugs for treating hypertension and atherosclerosis in China [7,8], was lower than that without the medicine [9]. The mechanism of the calcium channel blockers against SFTS virus seems to be an inhibition of cellular internalization of adherent viral particles and genome replication/transcription [9,10]. While many other small molecule inhibitors against the SFTS virus have been reported and some of them were examined in animal models [11,12], none of them have been administrated into SFTS patients.

The genome of the SFTS virus encodes four proteins, nuclear protein, glycoprotein (GP), RNA-dependent RNA polymerase, and a non-structural protein. GP is the viral envelope protein and is cleaved into Gn and Gc (Gn/Gc) during synthesis. While roles of Gn/Gc proteins have not been well explored, based on homology of amino acid sequence of the protein among viruses of *Phenuiviridae*, Gn and Gc seem to play a role in cell attachment and membrane fusion, respectively [13,14]. Activities of low-pH-dependent cellular membrane fusion among infected cells or Gn/Gc-expressing cells are affected by mutations within Gc [15,16]. There has been no report on a pathological function(s) of Gn/Gc in SFTS disease.

Monoclonal antibodies (mAbs) and immunoglobulins against viruses are in general expected to show therapeutic effects through inhibition of viral infection, enhancement of phagocytosis by opsonization, destruction of infected cells, and others [17], and have also been developed against severe diseases such as ebola virus disease, which shows quite high CFRs (e.g., 90%) [18]. In SFTS treatment, there are only a few reports involving the development of mAbs/immunoglobulins against the SFTS virus. Human mAbs MAb4-5 and Ab10 clones [19,20] and polyclonal sera [21] were established from patients recovered from SFTS and examined in vitro and in vivo. Both the two mAbs recognized Gn protein (K_D_ values 25.9 nM for MAb4-5 and 104 pM for Ab10), one of the envelope proteins of SFTS virus, and had neutralizing activities against the virus (approximately 80% inhibition at 5 μg/mL for both). The neutralizing activity of the human sera was 2000 [21]. Among these, Ab10 clone and the sera were shown to have therapeutic effects in IFNAR^-/-^ mice [20,21]. Unfortunately, due to a lack of comprehensive analyses, it is unclear what essence(s) are important for mAbs and sera as agents to show therapeutic effects.

In the present study, to reveal the potential of mAbs as therapeutic agents in SFTS treatment, we characterized mouse and human mAbs specific to the envelope proteins of the SFTS virus, Gn and Gc, in in vitro and in vivo experiments. The mAbs characterized were prepared in the present and previous studies, obtained commercially, or made based on publicized studies. Some mAbs showing high affinities and/or neutralizing activities were examined for therapeutic effects in an SFTS animal model. Data obtained will be useful in further development of therapeutic mAbs for SFTS treatment and in profound understanding of SFTS envelope protein roles.

## 2. Materials and Methods

### 2.1. Cells and Control Antibodies

Vero cells (ATCC CCL-81, ATCC, Manassas, VA, USA) were cultured in DMEM (Sigma-Aldrich, St. Louis, MO, USA) supplemented with 5% heat-inactivated foetal calf serum (FCS) and antibiotics (Gibco, Pen Strep, Grand Island, NY, USA). Upon inoculation, Vero cells were maintained in DMEM with 2% FCS (DMEM-2FCS) and antibiotics. 293T cells, HuH-7 cells and NIH/3T3 cells were cultured in DMEM (Sigma) supplemented with 10% FCS and antibiotics. BTI-TN-5B1-4 (Tn5) insect cells (Thermo Fisher Scientific, Waltham, MA, USA) were cultured in EX-CELL 405 Serum-Free Medium for Insect Cells (Sigma) supplemented with antibiotics (Gibco) at 28 °C. Normal human IgG, mouse IgG1 isotype control, and anti-*Strep*-Tag II monoclonal antibody were purchased from FUJIFILM Wako Pure Chemical Corp. (Tokyo, Japan), R&D Systems (Minneapolis, MN, USA), and IBA-Lifesciences (Göttingen, Germany), respectively.

### 2.2. SFTS Virus-Infected Cell Lysate

SFTS virus SPL030 strain was prepared as described previously [15]. Lysates of SFTS virus-infected and virus-uninfected HuH-7 cells were prepared as described previously [22].

### 2.3. Recombinant Proteins

For the preparation of a soluble form of the SFTS virus Gn, cDNA encoding amino acid position 1 to 429 of SPL030 strain GP fused with strep tag and HiBiT sequence (STH) at the C-terminus was cloned into pCAGGS plasmid [23]. For Gc, cDNA encoding amino acid position 563 to 996 fused with murine Ig kappa signal peptide at the N-terminus and STH at the C-terminus was cloned into pCAGGS plasmid. Expi293 Expression System (Thermo Fisher Scientific) was used with pCAGGS plasmids to produce soluble Gn (SFTSVGnSTH) and Gc (SFTSVGcSTH) proteins according to the manufacturer’s protocol. From culture supernatants, Gn and Gc proteins were purified by using Strep-Tactin Superflow high capacity column (IBA Lifesciences). Buffers were exchanged to PBS by using PD-10 column (Amersham Biosciences, Amersham, United Kingdom).

Soluble forms of the SFTS virus Gn fused with Fc portion of mouse IgG2a at the C-terminus were prepared as follows: cDNAs encoding amino acid position 1 to 429 or 1 to 340 of SPL030 strain GP were cloned into pCAGGS plasmid containing mouse IgG2a Fc cDNA. Two resultant plasmids and a control plasmid (no insertion) for Gn429mFc, Gn340mFc, and IgkmG2aFc proteins were transfected with TransIT LT1 transfection reagent (Takara Bio, Shiga, Japan) into 293T cells cultured in Opti-MEM culture medium (Thermo Fisher Scientific). Three days post transfection, supernatants were harvested and mixed with Protein G Sepharose 4 Fast Flow (Cytiva, Marlborough, MA, USA) to purify Fc-containing proteins.

To produce a soluble form of the SFTS virus Gn in a baculovirus expression system, cDNA encoding amino acid position 21 to 451 of SPL030 strain Gn fused with Honey Bee Mellitin secretion signal at the N-terminus and 8 × histidine at the C-terminus was cloned into pAcYM1 [24]. Recombinant baculoviruses were produced by using the plasmid as described previously [25] and purified SFTS virus Gn protein (SFTSVGn8His) was prepared as described previously [26].

### 2.4. Human mAbs

Monoclonal antibodies from the SFTS patient were established as previously described [27]. In brief, peripheral blood mononuclear cells (PBMCs) of SFTS patients were stained with the recombinant SFTSVGn8His fluorescent probes in DMEM supplemented with 2% FCS for 30 min at room temperature. This was followed by staining with IgG-FITC (G18-145, BD Biosciences, Tokyo, Japan), CD2-BV510 (RPA-2.10, BioLegend, San Diego, CA, USA), CD4-BV510 (RPA-T4, BioLegend), IgD-BV510 (IA6-2, BD Biosciences), CD10-BV510 (HI10a, BioLegend), CD14-BV510 (M5E2, BioLegend), CD27-Alexa700 (O323, BD Biosciences), CD19-BV786 (HIB19, BD Biosciences), LIVE/DEAD Fixable Aqua Dead Cell Stain kit (Thermo Fisher Scientific). Gn-binding IgG memory B cells were sorted into 96-well plates at one cell per well using FACS AriaIII (BD Biosciences). The VH/VL genes were PCR amplified and cloned into the expression vectors with human IgG1 heavy chain and kappa/lambda light chain. Pairs of heavy and light chain vectors were transfected into Expi293F cells according to the manufacturer’s instructions (Thermo Fisher Scientific). Thereafter, antibodies were purified from the culture supernatant using a protein G column (Thermo Fisher Scientific).

A human mAb Ab10 clone specific to the SFTS virus Gn reported by Kim et al. [20] was prepared based on the publication. A human mAb Ab3 clone specific to SFTS virus Gc was prepared based on US Patent information (#10947299). To obtain purified mAbs, the Expi293 Expression System was used with plasmids encoding heavy and light chains of the mAbs according to the manufacturer’s protocol, and immunoglobulins were purified with HiTrap™ Protein G HP Columns (Cytiva).

### 2.5. Mouse mAbs

NIH/3T3 cells were transfected with an SFTS virus Gn/Gc expression plasmid (pC030GP) [28] by using TransIT LT1, then used for immunization of BALB/c mouse to obtain antibody-producing hybridoma. Screening of hybridoma was performed with ELISA with SFTS virus-infected and virus-uninfected cell lysates (described above) as antigens. SFTSVGnSTH and SFTSVGcSTH proteins (described above) were used for immunization of GANP^®^ mice to obtain antibody-producing hybridoma and screening of hybridoma was performed with ELISA with each protein as antigens (Transgenic Corp., Hyogo, Japan). To obtain purified mAbs, hybridoma were cultured in Hybridoma Serum-Free Medium (FUJIFILM Wako Pure Chemical Corp.) and HiTrap™ Protein G HP Columns (Cytiva) were used. Isotypes of mouse mAbs obtained were determined with IsoStrip™ Mouse Monoclonal Antibody Isotyping Kit (Roche, Basel, Switzerland). Commercially available mouse mAbs specific to SFTS virus Gn and Gc were purchased from Immune Technology Corp. (New York, NY, USA). Two mAbs, C3A11 and C6C1 clones, were described previously [29].

### 2.6. Neutralization Tests

Replication-defective vesicular stomatitis virus vector (VSVΔG) encoding green fluorescent protein (GFP) and carrying SFTS virus Gn/Gc as envelope proteins [30] was used as a target virus. To prepare the recombinant vector, 293T cells were transfected with pC030GP [28] and, at 24 h post transfection, further infected with the vector trans-complemented with VSV G protein. The following day, culture supernatant containing the SFTS virus Gn/Gc-carrying vector was harvested and centrifuged to remove cell debris. The supernatant was mixed with anti-VSV G mAb I1 (8G5F11, Kerafast, Boston, MA, USA) to neutralize residual VSV G-carrying vector and then stored at −80 °C until use. The vector was mixed with mAbs at indicated concentrations and incubated for 1 h at room temperature. Vero cell monolayers seeded in 96-multiple well plates were inoculated with the vector-mAb mixture and incubated for 18 h at 37 °C. Images of inoculated cells were made under a BZ-X710 fluorescence microscope (KEYENCE, Osaka, Japan), and numbers of GFP-expressing cells were counted with ImageJ software.

To test whether neutralizing mAbs had different inhibitory effects on VSVΔG carrying the SFTS virus Gn/Gc, mAbs, mouse IgG1 isotype control, and normal human IgG were used at 16 µg/mL before (the vector was treated with antibodies then inoculated onto Vero cells) and after (the vector was inoculated first onto Vero cells in the absence of antibodies then cultured with media containing antibodies) viral adsorption to cells. When antibodies were used before adsorption, the vector was treated with antibodies for 1 h on ice then inoculated onto Vero cells in a cold room. After 1 h incubation, inocula were removed and cells were washed with cold DMEM-2FCS three times then cultured in the absence of antibodies. When antibodies were used after adsorption, the vector was inoculated onto Vero cells without antibodies in a cold room. After 1 h incubation, inocula were removed and cells were washed with cold DMEM-2FCS three times then cultured in the presence of antibodies. Cell culture and reporter analyses were performed as described in the previous paragraph.

### 2.7. Therapeutic Effects

Mice lacking type I interferon receptor (IFNAR^-/-^) [31] subcutaneously inoculated with 10^2^ 50% tissue culture infectious dose (TCID_50_) of SFTS virus SLP030 strain were used as a fatal animal model of SFTS. To assess the therapeutic effects of mAbs, the inoculated mice (3 to 6 mice per group) were intraperitoneally injected with indicated doses of mAbs daily between 1 day-post inoculation (dpi) and 6 dpi. Mice were observed daily up to 14 dpi.

### 2.8. Competition ELISA

SFTSVGnSTH and SFTSVGcSTH proteins (described above) were used as antigens to coat Nunc MaxiSorp™ flat-bottom (Thermo Fisher Scientific). After blocking with skim milk, antigens were first incubated with unlabeled mAbs at indicated concentrations for 30 m, followed by the addition of biotin-labeled mAbs (Biotin Labeling Kit—NH_2_, DOJINDO Laboratories, Kumamoto, Japan) at optimized dilutions and incubation for 30 m. After washing, plates were incubated with peroxidase-conjugated streptavidin (Proteintech, Rosemont, IL, USA) at 1:2000 for 30 m. Colorization was performed with ABTS tablet (Roche) and optical densities at 405 nm were measured (iMark microplate reader, Bio-Rad, Hercules, CA, USA).

### 2.9. ELISA with Fc-Tagged Gn

Gn429mFc, Gn340mFc, and IgkmG2aFc (control) proteins were used as antigens to coat Nunc MaxiSorp™ flat-bottom. After blocking with skim milk, antigens were incubated with biotin-labeled mAbs followed by peroxidase-conjugated streptavidin or directly incubated with goat anti-mouse IgG (H+L) secondary antibody, HRP (Thermo Fisher Scientific). Colorization and optical density measurement were performed as described above. The ratios of densities with peroxidase-conjugated streptavidin to those with anti-mouse IgG (H+L) secondary antibody were calculated.

### 2.10. Ethical Statements

PBMCs were obtained from SFTS-recovered patients for the preparation of human mAbs. Protocols and procedures were approved by the research ethics committee of the National Institute of Infectious Diseases for the use of human subjects (no. 539). Experiments with animals were performed in strict accordance with the Animal Experimentation Guidelines of the National Institute of Infectious Diseases. The protocol of animal experiments were approved by the Institutional Animal Care and Use Committee of the National Institute of Infectious Diseases (nos. 117143 and 118121).

### 2.11. Statistics

Survival curves were plotted using the Kaplan-Meier method and effects of mAb administration was evaluated using a log-rank test. A *p*-value less than 0.05 was considered significant statistically. Student’s *t*-test was used to determine the significance of the difference between the means of two groups.

## 3. Results

### 3.1. mAbs Specific to SFTS Virus Gn and Gc

Four human mAbs (clones M1-B8, M1-D1, M1-E1, and M1-E5) were obtained by using PBMCs from an SFTS patient and SFTS virus Gn protein (for details see Materials and Methods). Three murine mAbs (clones 5D12, 6D12, and 8E9) were established from BALB/c mice immunized with the SFTS virus Gn/Gc-expressing cells, and all the three recognized Gc protein in ELISA. Murine mAbs to Gn (No. 2, 4, 5, and 11) and to Gc (No. 16, 22, 23, 31, 33, and 40) were established from GANP^®^ mice immunized with SFTS virus Gn and Gc proteins. Human mAbs MAb4-5 (Gn; [19]), Ab10 (Gn; [20]), and Ab3 (Gc; patent #10947299) were produced based on the publicized information. These mAbs, together with commercially available murine mAbs (4M2, 4M3, 4M5, 4M6, 4M7, and 4M8 for Gn and 5M1, 5M3, 5M5, and 5M9 for Gc), were examined to measure half maximal effective concentrations (EC_50_) in ELISA and half maximal inhibitory concentrations (IC_50_) in a neutralization test (NT). The maximum concentration of mAbs examined were 16µg/mL. Results are shown in Table 1 and Table 2, Figure 1, and Appendix A. EC_50_ in ELISA of the anti-Gn and the anti-Gc mAbs were between 0.016 and 2.2 μg/mL. In NT, mAb clones which had IC_50_ of less than 16 μg/mL were 4M5 (anti-Gn, 0.17 μg/mL), Ab10 (anti-Gn, 0.35 μg/mL), No. 22 (anti-Gc, 0.66 μg/mL), No. 23 (anti-Gc, 1.1 μg/mL), 5M5 (anti-Gc, 3.9 μg/mL), 5M9 (anti-Gc, 2.1 μg/mL), C6C1 (anti-Gc, 1.8 μg/mL), and Ab3 (anti-Gc, 0.040 μg/mL).

### 3.2. Therapeutic Effects of Selected mAbs in an SFTS Animal Model

Based on the results obtained in ELISA, NT, and/or mAb usability, anti-Gn mAb clones (4M5, 4M6, M1-B8, M1-D1, M1-E1, M1-E5, Ab10, and MAb4-5) and anti-Gc mAb clones (No. 22, 5D12, 6D12, 8E9, C3A11, C6C1, and Ab3) were tested with regard to their therapeutic effects with SFTS virus-infected IFNAR^-/-^ mice. mAb doses used were 0.02 mg/day for 4M5 and 4M6 (control for 4M5) clones and 1 mg/day for the others. As shown in Figure 2a, all mock-treated mice, which were infected with the SFTS virus and then injected with PBS instead of mAb, started body weight loss at 3 dpi or later and died at 6 dpi or 7 dpi. Injection of normal human IgG (1 mg/day) had no effects on survival and body weight changes of mice (Figure 2a). Survived mice were observed within 4M5, M1-E5, and Ab10 mAb-treated groups: 4M5-injection resulted in body weight loss between 3 dpi and 8 dpi, but all mice used survived (Figure 2b), M1-E5-injection delayed mice death and one mouse survived among five mice used (Figure 2a), and Ab10-injection resulted in no apparent body weight loss and all mice used survived (Figure 2a). In experiments with reduced doses of Ab10 mAb, no apparent body weight loss nor death was observed with 0.1 and 0.01 mg/day injection, however, 0.001 mg/day of Ab10 resulted in body weight loss but no death (Figure 2c). Injection of 0.0001 mg/day of Ab10 had no effects on body weight loss and death when compared with PBS- or normal human IgG-injection. The other mAb injection resulted in body weight loss and no mice injected survived until 9 dpi (Appendix A). Therapeutic effects of the mAbs are summarized in Table 1 and Table 2.

### 3.3. Epitopes of Anti-Gn mAbs

Epitopes recognized by anti-Gn mAbs, Ab10, 4M5, and M1-E5 were examined with SFTS virus Gn protein in competition ELISA (see Materials and Methods for details). Among the three mAbs, as shown in Table 3 and Appendix A, the binding of biotinylated mAbs were reduced by unlabeled respective mAbs but not the others dose-dependently. Any other anti-Gn mAbs did not inhibit biotinylated Ab10 mAb (Appendix A). A reduction in biotinylated 4M5 mAb binding was observed with 4M6 and 4M8, and an apparent reduction in biotinylated M1-E5 mAb binding was observed with M1-B8, M1-D1, and M1-E1 mAbs (Appendix A).

Epitopes of Ab10, 4M5, and M1-E5 mAbs were further examined in ELISA with antigens including truncated Gn proteins, which were consisted with amino acid position 1 to 429 (Gn429mFc) or 1 to 340 (Gn340mFc) of the SFTS virus Gn. Both of the Ab10 and M1-E5 mAbs showed similar reactivity to Gn429mFc and Gn340mFc, however, 4M5 mAb only showed a positive signal to Gn429mFc (Figure 3).

SFTS virus Gn and Gc proteins were used in Western blotting under reduced conditions. As shown in Figure 4, both of 4M5 and M1-E5 mAbs detected Gn, but not the Gc protein, suggesting the mAbs recognized linear epitopes among the SFTS virus Gn protein.

### 3.4. Epitopes of Anti-Gc mAbs

Epitopes recognized by anti-Gc mAbs, No. 22, C6C1, and Ab3 were examined with the SFTS virus Gc protein in competition ELISA. Among the three mAbs, the binding of biotinylated mAbs were reduced by unlabeled respective mAbs dose-dependently (Table 4), indicating that there were three or more neutralizing epitopes among the Gc protein of the SFTS virus. The binding of biotinylated C6C1 and Ab3 mAbs were not reduced by the others, while those of No. 22 mAb were reduced by No. 23, 5M9, and C3A11 mAbs (Appendix A).

### 3.5. A Different Property of Neutralizing Anti-Gn/Gc mAbs

To speculate which steps during viral infection were affected by neutralizing anti-Gn mAbs (Ab10, 4M5, and M1-E5) and anti-Gc mAbs (No. 22, C6C1, and Ab3), mAbs were used at two different time points in NT (see Materials and Methods for details) and results were compared between the time points. Inhibitory effects of Ab10 and M1-E5 on VSVΔG carrying the SFTS virus Gn/Gc infection were reduced when the mAbs were used after viral adsorption to cells, in comparison to those when used before (Appendix A). Such a reduction in inhibitory effects was not observed for other neutralizing mAbs examined (Appendix A).

## 4. Discussion

Among 32 mAbs recognizing envelope proteins of SFTS virus Gn/Gc, three clones, Ab10, 4M5, and M1-E5, showed therapeutic effects in murine model of SFTS. Ab10 is a human mAb which has been reported in Kim’s study [20], in which the authors examined therapeutic effects at a dose of 600 µg/day x 4 days, and was most effective in the present study. We examined fewer doses for the mAb, and found that a dose of 1 µg/day x 6 days was effective as a treatment in the SFTS murine model. The remaining two mAbs were apparently less effective than Ab10, but were novel ones showing therapeutic effects against SFTS. One of the remaining two, 4M5, is a commercially available murine mAb (Immune Technology Corp) and was effective at a dose of 20 µg/day x 6 days. The last was human mAb M1-E5, whose therapeutic effect was considerably weak when compared with Ab10 and 4M5, but showed life-extending effects at a dose of 1 mg/day x 6 days. All of these three mAb clones recognized Gn proteins and showed neutralizing effects with calculable IC_50_ values or reduced viral infectivity to nearly 50% at 4 or 16 µg/mL. In contrast, any of anti-Gc mAbs examined did not show therapeutic effects at a dose of 1 mg/day x 6 days, whereas some anti-Gc mAbs showed comparable to or higher neutralizing activities than anti-Gn mAbs. From these results, we propose that to obtain promising therapeutic mAb candidates for SFTS treatment effectively, mAbs recognizing Gn protein of the SFTS virus and having neutralizing activities against the virus should be picked up during selection. Because Ab10 and 4M5 did not interfere their binding to Gn protein with each other, and the EC_50_ value of 4M5 was not so low (indicating low affinity), mAbs recognizing 4M5′s epitope with high affinities, if obtained, might become a promising therapeutic candidate such as Ab10. Because the number of mAbs with therapeutic effects were limited, it is unfortunately unclear whether isotypes and affinities to antigens of mAbs were important as essences to predict therapeutic effects against SFTS.

Kim et al. reported that the Ab10 clone recognizes a conformational epitope existing within C-terminal half of Gn [20]. Virus neutralization, competition ELISA, and ELISA with truncated antigens performed in the present study showed that 4M5 and M1-E5 clones recognize distinct epitopes which are also distinct from Ab10′s one, indicating that there are three or more epitopes within Gn mAb bindings to which led to virus neutralization and inhibition/delay of death of SFTS virus-infected mice. Interestingly, although bindings of 4M5 and M1-E5 to Gn were reduced by 4M6, M1-B8, M1-D1, or M1-E1, none of the latter four mAbs showed neutralizing activities and therapeutic effects. Therefore, it would not be a preferable strategy in which mAbs block 4M5- or M1-E5-binding would be selected for therapeutic mAb development.

Whereas analyses of the neutralization mechanism(s) of mAbs are not the original goal of the present study, it was observed that the inhibitory effects of two anti-Gn mAbs, Ab10 and M1-E5, on viral infection were decreased when the mAbs were used after the viral attachment to cells, suggesting that the mAbs might block the attaching step of SFTS virus infection. This is consistent with the previous work with SFTS virus-related viruses reporting a role of the Gn protein during infection [13]. Interestingly, anti-Gn neutralizing mAb 4M5 seemed not to block the attaching step. The mAb recognized the most C-terminal part within the extracellular region of Gn and, therefore, might affect the interaction between Gn and Gc, while analyses of the interaction, to our knowledge, has never been reported in SFTS virus. It was also observed that the inhibitory effects of anti-Gc neutralizing mAbs, No. 22, Ab3, and C6C1, were not lessened when the mAbs were used after the viral attachment to cells, suggesting that the mAbs might block a step(s) following SFTS virus attachment to cells, e.g., fusion between virus and cell membranes [14]. In any case, identification of the SFTS virus receptor(s), clarification of Gn-Gc interaction including spatial competition and virus-cell membrane fusion mechanism will lead to elucidation of neutralization mechanisms of these Gn/Gc mAbs.

Results of neutralizing activities and therapeutic effects of mAbs against SFTS virus revealed several points which should be resolved in future studies to understand SFTS pathogenesis more profoundly and enhance development of therapeutic agents. First is the role of the Gn protein of SFTS virus, especially in SFTS murine models. The anti-Gc mAb, Ab3, showed the strongest neutralizing activity among 32 mAbs examined in the present study, but did not show any therapeutic effects such as inhibition or delay of body weight loss and death upon SFTS virus infection. Although having less neutralizing activities than Ab3, some anti-Gn mAbs apparently showed therapeutic effects. Therefore, we speculate that there might be an unknown role(s) of Gn in vivo, for example, stimulation of some immune cells by binding to their surface, which eventually might lead to hyper cytokine production and could be inhibited by some anti-Gn mAbs. Second is the different involvement of Gn/Gc proteins of authentic SFTS virus particles and of VSV pseudotypes in infection to cells. VSV pseudotypes are a safe and convenient tool due to their replication incompetency and reporter expression, however there is currently no evidence which supports the identical or similar functions of Gn/Gc between authentic SFTS virus particles and VSV pseudotypes. Contribution of Gn might be greater than that of Gc in authentic viral particles and vice versa in VSV particles. Supportive of this speculation is that maximum inhibitory effects of anti-Gn mAbs were approximately 40% of control, while those of anti-Gc mAbs were 40% or less (Figure 1). In this regard, SFTS virus-based replication-incompetent particles, e.g., iVLP [28], might be usable instead of VSV-based pseudotypes, because iVLP has nearly same composition with the authentic SFTS virus.

Our results in the present study propose an effective strategy to obtain promising therapeutic mAb candidates for SFTS treatment, and a necessity to reveal precise roles of the SFTS virus Gn/Gc proteins.

## Figures and Tables

**Figure 1 viruses-14-01665-f001:**
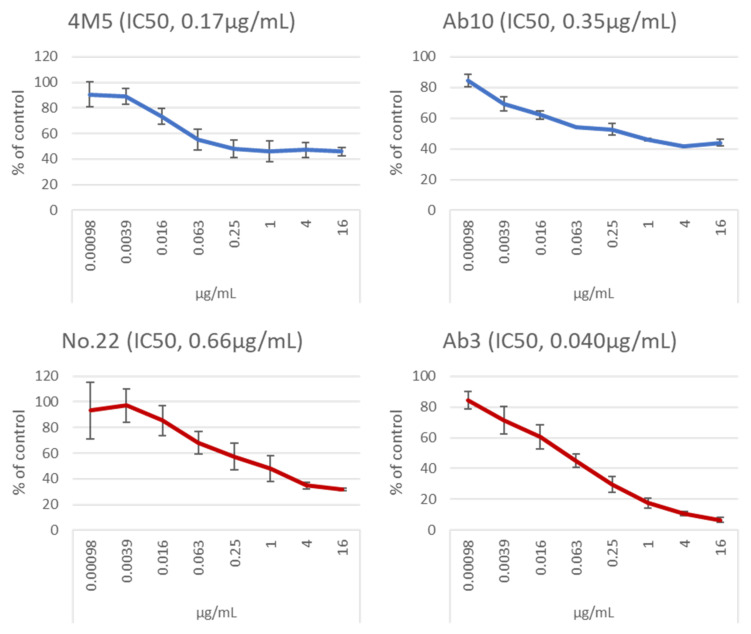
Inhibitory effects of two anti-Gn mAbs (4M5 and Ab10) and two anti-Gc mAbs (No. 22 and Ab3) against replication-defective vesicular stomatitis virus vector carrying SFTS virus Gn/Gc as envelope proteins. The *x*-axis indicates concentration of the mAb used to treat virus and the *y*-axis indicates reporter-positive cell number percentages against control (mAb 0 µg/mL). Experiments were performed at triplicates. Data are the means ± standard deviations.

**Figure 2 viruses-14-01665-f002:**
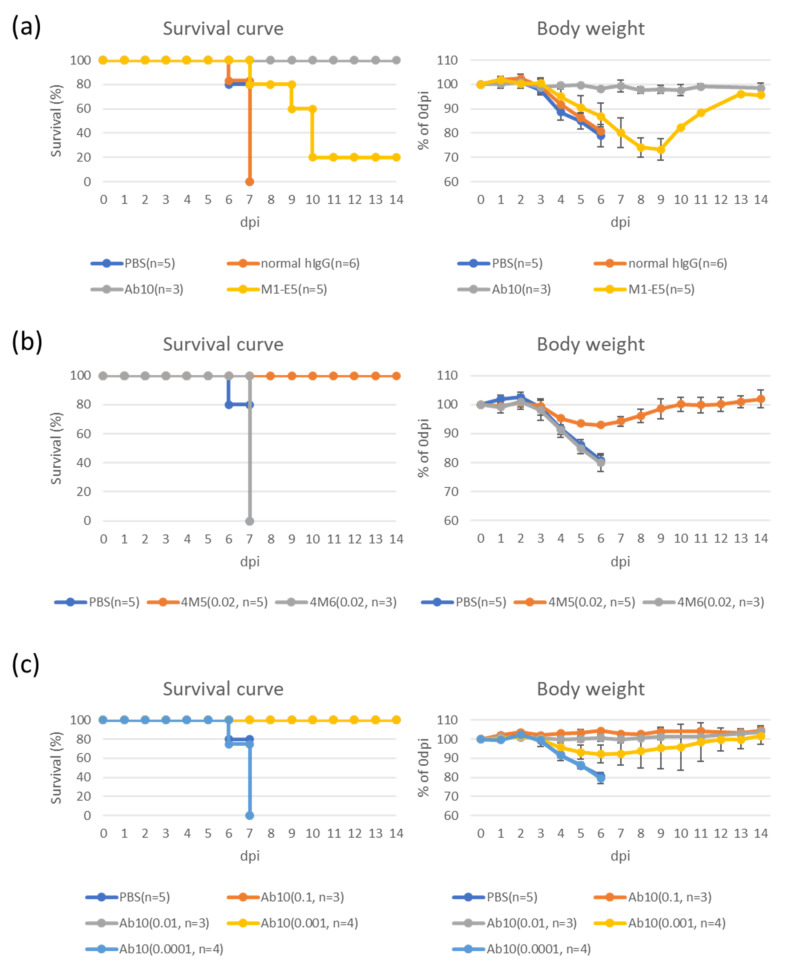
Therapeutic effects of mAbs in a fatal animal model of SFTS. IFNAR^-/-^ mice infected with SFTS virus were intraperitoneally injected with 1 mg (**a**) or indicated doses (mg) in parentheses (**b**) and (**c**) of mAbs daily between 1 dpi and 6 dpi. The parentheses also include numbers of mice used (**a**–**c**). Mice were observed for their survival/death (left, Kaplan-Meier) and body weight (right) daily up to 14dpi. On the right, body weight percentages against 0 dpi are shown. Data of body weight are shown in means ± standard deviations.

**Figure 3 viruses-14-01665-f003:**
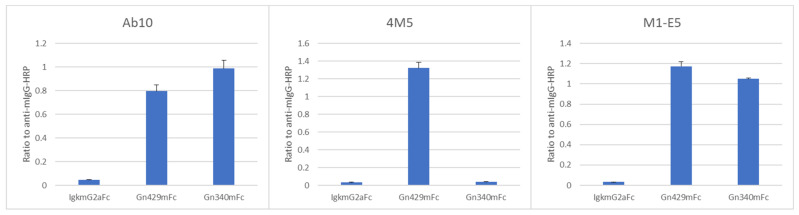
ELISA with truncated Gn proteins. Truncated Gn proteins were detected by anti-mouse IgG labeled with HRP (for normalization) or by biotinylated mAbs with streptavidin HRP. Experiments were performed at triplicates and data are means ± standard deviations of ratios of mAb-absorbance to anti-mouse IgG-absorbance.

**Figure 4 viruses-14-01665-f004:**
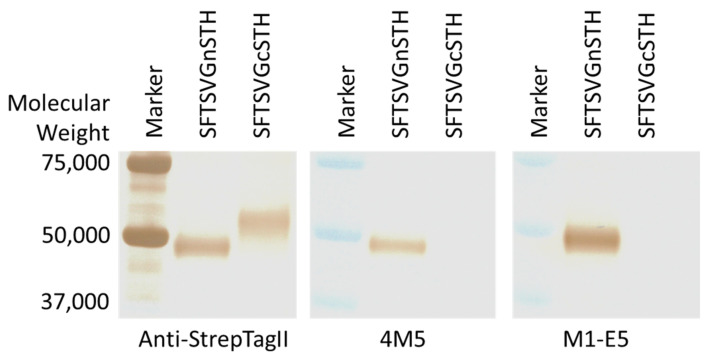
Western blotting with SFTS virus Gn and Gc proteins under reduced condition. Three mAbs, anti- Strep Tag II, 4M5, and M1-E5, were used as primary antibodies.

**Table 1 viruses-14-01665-t001:** mAbs specific to SFTS virus Gn protein.

Clones	Isotypes	ELISA EC_50_, µg/mL	NT IC_50_, µg/mL	Therapeutic Effects with Doses, mg/day	References
No. 2	Mouse IgG1	0.14	16<	ND ^1^	Present study
No. 4	Mouse IgG1	0.090	16<	ND	Present study
No. 5	Mouse IgG1	0.11	16<	ND	Present study
No. 11	Mouse IgG1	0.10	16<	ND	Present study
4M2	Mouse IgG2b	0.12	16<	ND	Immune Technology Corp.
4M3	Mouse IgG2a	0.046	16<	ND	Immune Technology Corp.
4M5	Mouse IgG1	0.16	0.17	Y ^2^ at 0.02	Immune Technology Corp.
4M6	Mouse IgG1	0.067	16<	N ^3^ at 0.02	Immune Technology Corp.
4M7	Mouse IgG1	0.062	16<	ND	Immune Technology Corp.
4M8	Mouse IgG2b	0.016	16<	ND	Immune Technology Corp.
M1-B8	Human IgG1	0.056	16<	N at 1	Present study
M1-D1	Human IgG1	0.061	16<	N at 1	Present study
M1-E1	Human IgG1	0.051	16<	N at 1	Present study
M1-E5	Human IgG1	0.046	16<	Y at 1	Present study
Ab10	Human IgG1	0.027	0.35	Y at 1, 0.1, 0.01, 0.001N at 0.0001	[20]
MAb4-5	Human IgG1	0.48	16<	N at 1	[19]

^1^ ND, not done. ^2^ Y, positive results observed. ^3^ N, positive results not observed.

**Table 2 viruses-14-01665-t002:** mAbs specific to SFTS virus Gc protein.

Clones	Isotypes	ELISA EC_50_, µg/mL	NT IC_50_, µg/mL	Therapeutic Effects with Doses, mg/day	References
No. 16	Mouse IgG1	0.18	16<	ND ^1^	Present study
No. 22	Mouse IgG2b	0.077	0.66	N ^2^ at 1	Present study
No. 23	Mouse IgG1	0.96	1.1	ND	Present study
No. 31	Mouse IgG1	0.11	16<	ND	Present study
No. 33	Mouse IgG1	0.61	16<	ND	Present study
No. 40	Mouse IgG1	0.49	16<	ND	Present study
5D12	Mouse IgG1	0.14	16<	N at 1	Present study
6D12	Mouse IgG2a	0.50	16<	N at 1	Present study
8E9	Mouse IgG1	2.2	16<	N at 1	Present study
5M1	Mouse IgG1	0.22	16<	ND	Immune Technology Corp.
5M3	Mouse IgG1	0.045	16<	ND	Immune Technology Corp.
5M5	Mouse IgG1	0.094	3.9	ND	Immune Technology Corp.
5M9	Mouse IgG1	0.037	2.1	ND	Immune Technology Corp.
C3A11	Mouse IgG2a	0.048	16<	N at 1	[29]
C6C1	Mouse IgG1	0.053	1.8	N at 1	[29]
Ab3	Human IgG1	0.054	0.040	N at 1	Patent (#10947299)

^1^ ND, not done. ^2^ N, positive results not observed.

**Table 3 viruses-14-01665-t003:** Competition ELISA with anti-Gn mAbs ^1^.

Competitors ^2^	Binding of Biotinylated Ab10	Binding of Biotinylated 4M5	Binding of Biotinylated M1-E5
Ab10	Reduced	Not reduced	Not reduced
4M5, 4M6, 4M8	Not reduced	Reduced	Not reduced
M1-B8, M1-D1, M1-E1, M1-E5	Not reduced	Not reduced	Reduced ^1^

^1^ Detailed data are shown in Appendix A. Effects of the others on the biotinylated mAb binding are shown in Appendix A. ^2^ No. 2, No. 4, No. 5, No. 11, 4M2, 4M3, 4M7, and MAb4-5 did not reduce any biotinylated anti-Gn mAbs examined significantly.

**Table 4 viruses-14-01665-t004:** Competition ELISA with anti-Gc mAbs ^1^.

Competitors ^2^	Binding of Biotinylated No. 22	Binding of Biotinylated C6C1	Binding of Biotinylated Ab3
No. 22, No. 23, 5M9, C3A11	Reduced	Not reduced	Not reduced
C6C1	Not reduced	Reduced	Not reduced
Ab3	Not reduced	Not reduced	Reduced ^1^

^1^ Detailed data are shown in Appendix A. Effects of the others on the biotinylated mAb binding are shown in Appendix A. ^2^ No. 16, No. 31, No. 33, No. 40, 5D12, 6D12, 8E9, 5M1, 5M3, and 5M5 did not reduce any biotinylated anti-Gc mAbs examined significantly.

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
