# Peer review of "Neutralizing mAbs against SFTS Virus Gn Protein Show Strong Therapeutic Effects in an SFTS Animal Model"

_viruses, 2022, doi:10.3390/v14081665_

Round 1
Reviewer 1 Report
In this study, the authors found that mAbs have therapeutic effects in murine model of SFTS. Therapeutic activities were observed among anti-Gn, but not anti-Gc, mAbs with neutralizing activities. These results propose an effective strategy to obtain promising therapeutic mAb candidates for SFTS treatment and a necessity to reveal precise roles of SFTS virus Gn/Gc proteins. The study design and presentation are detailed. The data presented are generally strong, and appear convincing. The work is based on a very strong foundation with clear and robust data, but would benefit with further experiments to help strengthen the main conclusions and to better understand.
Major comments:
1. All charts are missing x and y axes, which are not normative.
2. Figure1, Inhibitory effects of mAbs should be presented in curve with EC50 presented.
3. Figure2, the author should indicate how many mice were used in each group.
4. For detecting the epitopes of anti-Gn mAbs, site-mutants of Gn should be generated for detecting, which could clearly help the author find the recognize site of these anti-Gn maAbs. Using competition ELISA and truncated proteins could only know the approximate range.
5. The author should detect the antibody affinity of these mAbs, because Antibody affinity is defined as strength of the binding interaction between antigen and antibody.
Minor comments:
1. Some language errors need to be corrected in this manuscript. Such as “in vivo” and “in vitro” should be italicized.
Reviewer 2 Report
In this experiment, the authors collected a large number of monoclonal antibodies against GP of SFTSV and carefully analyzed them to demonstrate their potential therapeutic application. The experiment is carefully planned and carried out towards the goal. The author has done extensive experiments with many antibodies and may need to make a few more changes to explain the results.
1. The results of the competitive inhibition assay in Fig. 3 are difficult to understand. If there was a table that also included the results in the Supplemental data and Fig. 3, it will be convenient for readers.
2. In this experiment, authors are conducting a competitive test using Gn and Gc antigens expressed separately. As a result, the neutralizing epitopes of Gn and Gc are described as different epitopes. However, there is a possibility that Gn and Gc will also compete with each other. Spatial competition on the GP cannot be detected in this experiment. Please add some helpful observations or considerations about Gn and Gc interaction.
3. Please comment on the neutralization of MAbs in vitro if you know. Do MAbs inhibit cell fusion activity or binding to target cells? I think this information will reinforce the discussion about the therapeutic effects in vivo.
4. In Fig. 5, please show the size of the molecular weight marker.
5. You need a space between the units and the number. e.g.) 5nM -> 5 nM
Round 2
Reviewer 1 Report
The authors addressed many of my previous criticisms thoroughly and elegantly the crucial major points of the revision. This is a very nice work and I would like to recommend it to be accepted.